# Mechanically Enhanced Detoxification of Chemical Warfare Agent Simulants by a Two-Dimensional Piezoresponsive Metal–Organic Framework

**DOI:** 10.3390/nano14070559

**Published:** 2024-03-22

**Authors:** Yuyang Liu, Shiyin Zhao, Yujiao Li, Jian Huang, Xuheng Yang, Jianfang Wang, Cheng-an Tao

**Affiliations:** College of Science, National University of Defense Technology, Changsha 430083, China; liuyuyang0412@163.com (Y.L.); zhaoshiyin21@nudt.edu.cn (S.Z.); liyujiao15@nudt.edu.cn (Y.L.); huangjian2015@nudt.edu.cn (J.H.); ouyangxuheng@163.com (X.Y.)

**Keywords:** piezoresponsive metal–organic frameworks, chemical warfare agent detoxification, mechanical energy harvesting

## Abstract

Chemical warfare agents (CWAs) refer to toxic chemical substances used in warfare. Recently, CWAs have been a critical threat for public safety due to their high toxicity. Metal–organic frameworks have exhibited great potential in protecting against CWAs due to their high crystallinity, stable structure, large specific surface area, high porosity, and adjustable structure. However, the metal clusters of most reported MOFs might be highly consumed when applied in CWA hydrolysis. Herein, we fabricated a two-dimensional piezoresponsive UiO-66-F_4_ and subjected it to CWA simulant dimethyl-4-nitrophenyl phosphate (DMNP) detoxification under sonic conditions. The results show that sonication can effectively enhance the removal performance under optimal conditions; the reaction rate constant *k* was upgraded 45% by sonication. Moreover, the first-principle calculation revealed that the band gap could be further widened with the application of mechanical stress, which was beneficial for the generation of ^1^O_2_, thus further upgrading the detoxification performance toward DMNP. This work demonstrated that mechanical vibration could be introduced to CWA protection, but promising applications are rarely reported.

## 1. Introduction

Chemical warfare agents (CWAs) refer to various chemical substances with severe toxicity used for war purposes, capable of poisoning or killing enemy humans, animals, and plants on a large scale. In recent years, CWAs have been one of the crucial means for terrorists and extremist organizations to enforce terrorist activities and war activities due to their large killing ability, wide working range, strong concealment, and low cost [1,2,3]. Consequently, due to their terrible toxicity, the quick detoxification of CWAs has become one of the critical demands in preventing CWA attacks [4,5]. To deal with CWAs, conventional methods include adsorption and chemical reaction methods [6,7,8]. The adsorption method presents good universality; however, the removal performance largely relies on the adsorption capacity of the adsorbents (e.g., activated carbon). Moreover, after reaching saturation, there is a large degree of desorption, which is potentially dangerous, thus hindering its further application in protective equipment [9,10,11]. The chemical reaction method exhibits the great advantage of permanent detoxification. Recently, most of the commonly used disinfectants have included oxidizing chlorinated disinfectants, alkaline disinfectants, etc. [12,13,14,15]. However, various problems also exist such as low efficiency, strong corrosivity, storage difficulties over long periods of time, and serious environmental pollution. Moreover, the consumption of these disinfectants is always huge; it is difficult to carry them in military operations, and it is also not easy to integrate them into military personal protective equipment. Therefore, developing new detoxification materials is urgently needed to guard against potential threats from chemical warfare agents [16,17,18].

In recent years, researchers have made great efforts in developing new detoxification materials, especially materials with catalytic properties, such as biological enzymes, anionic polyoxometalates, nano metal oxides, amphiphilic polyoxometalates, etc. These materials show a certain degree of detoxification ability; however, a variety of demerits limit their further promotion and application. For instance, biological enzymes exhibit high cost; the pore structure of metal oxides is relatively inferior; simultaneously, the surface chemical properties of metal oxides are usually ultrastable and are hard to modify, thus leading to the active adsorption ability of metal oxides to chemical warfare agents being insufficient [17,19,20,21]. To achieve the ideal detoxification effect, the designed material needs the following key properties. Firstly, the material needs an outstanding pore structure to provide more active sites and larger space for further modification. Secondly, the chemical structure of the material is adjustable and easy to modify, so that the bandgap width, affinity, and application range can be controlled. Moreover, the material needs to have the characteristic of utilizing external driving energy to enhance its reusability and be able utilize external driving forces to achieve and accelerate the catalytic process during detoxification; then, the crystals can be well preserved even after many cycles of use.

Metal–organic frameworks (MOFs) are porous crystalline materials assembled by the coordination of metal clusters and organic ligands [22,23,24,25]. They have outstanding advantages of high crystallinity, stable structure, large specific surface area, high porosity, and adjustable structure, which give them broad application prospects in various fields, such as adsorption, drug delivery, sensing, and photocatalysis [26,27,28,29]. MOFs are also regarded as some of the ideal detoxification materials for CWAs and have had significant outcomes in the field of detoxification, greatly improving detoxification efficiency and largely shortening the half-life of the CWAs. However, the active sites are easy to inactivate in the detoxifying process, which severely hinders their further application. External driving energy is urgently required to be employed to transfer to chemical energy, which can detoxify CWAs. Most reported works developed photocatalytic MOFs to harvest solar energy, which were applied in the degradation of pollutants and CWA simulants, for instance, PCN-222, Cu-TCPP, NU-1000, etc. [30,31,32,33]. However, the working performance of photoresponsive MOFs is largely limited by the weather and daytime duration. Mechanical energy is one of the most common forms of energy in nature, and piezocatalysis has been emerging as a promising alternative technology for organic pollutant removal, as it can harvest various mechanical energies including vibrations, wind, and water waves from the surrounding environment [34,35,36,37,38]. However, the mechanically enhanced detoxification of CWAs by piezoresponsive MOFs has rarely been reported [39].

Herein, we fabricated a two-dimensional UiO-66-F_4_ by the microwave-assisted hydrothermal method and subjected it to CWA simulant detoxification under mechanical vibration. The obtained UiO-66-F_4_ presented ultrathin nanosheets with a thickness of ~5 nm and a large specific surface area of 331.827 m^2^/g. It also showed outstanding piezoresponse capacity in an ultrasonic environment; the amplitude displacement of UiO-66-F_4_ could reach ~50 mV under a 10 V DC bias field. Consequently, it could effectively harvest mechanical energy and transfer it to chemical energy, thus effectively detoxifying the CWAs. Under optimal conditions, the UiO-66-F_4_ could detoxify almost all of the DMNP within 60 min, and the half-time of the DMNP was 10.58 min, which was much shorter than that under stirring conditions. The detoxification capacity was also excellent compared to other reported materials. Simultaneously, the reaction constant *k* was upgraded by 45% by the mechanical vibration. The first-principle calculation revealed the polarization behavior of the MOF crystals and the widening of the band gap under mechanical stress. This work demonstrated that CWAs could be mechanically detoxified by a piezoresponsive MOF, which provides great prospects for protection against CWAs of personnel in a toxic environment.

## 2. Materials and Methods

### 2.1. Chemicals

Tetrafluoroterephthalic acid (BDC-F_4_) was bought from Yanshen Chemistry (Jilin, China). Zirconium chloride (ZrCl_4_), methanol, absolute ethanol, N,Ndimethylformamide (DMF), and glacial acetic acid (HAc) were bought from Macklin (Shanghai, China). Thymolphthalein, sodium hydroxide (NaOH), zinc oxide (ZnO), and *N*-Ethylmorpholine were bought from Aladdin (Shanghai, China). Dimethyl-4-nitrophenyl phosphate (DMNP) was synthesized by our lab, and the chemical structure is displayed in Appendix A.

### 2.2. Characterizations

The morphology of the UiO-66-F_4_ was observed by field emission scanning electron microscopy (SEM, Zeiss Merlin, Oberkochen, Germany) with a working voltage of 10 kV and transmission electron microscopy (TEM, JEOL JEM F200, Tokyo Metropolitan, Japan) with a working voltage of 300 kV. The crystallinity was analyzed through a powder X-ray diffraction pattern which was recorded by a polycrystalline diffractometer (Rigaku MiniFlex 600, Tokyo, Japan) with Cu Kα radiation (λ = 0.154 nm). The Fourier transform infrared (FTIR) spectra were recorded by a PerkinElmer Spectrum-II spectrometer (Wellesley, MA, USA) with KBr pellets. The elements’ valence status was investigated through X-ray photoelectronic spectroscopy (XPS) which was conducted by a Thermo ESCALAB 250XI (Waltham, MA, USA) fitted with Al Kα radiation, and all the binding energy was referenced as the C 1s peak at 284.8 eV. The surface area and porosity were analyzed by Brunauer–Emmett–Teller (BET) and Horvath–Kawazoe (HK) methods, respectively, using a Quantachrome Autosorb IQ (Boynton Beach, FL, USA) instrument. The thickness and piezoresponse capacity were measured through atom force microscopy (AFM) and piezoresponse force microscopy (PFM) by a Bruker-Icon (Billerica, MA, USA) with a working probe (MESP, coefficient of elasticity 2 N/m, resonance frequency 75 Hz) and polarization voltage of 10 V, respectively. The size distribution and zeta potential were collected by a Nanotrac Wave II (Microtrac MRB, Clearwater, FL, USA). The ultraviolet and visible spectrum (UV-Vis) was measured by a UV-5200 (Metash, Shanghai, China) with a working wavelength range of 200–800 nm.

### 2.3. Assembly of UiO-66-F_4_

The UiO-66-F_4_ was assembled by a previously described method [38]. Typically, 85.9 mg BDC-F_4_ was added to 50 mL mixed solution (water/HAc = 30/20). After sonication for 5 min, 75 mg ZrCl_4_ was added, followed by continuous sonication for 10 min. Right after, the mixed solution was placed in a microwave oven and heated at 100 °C for 4 h. The obtained precipitates were washed by DMF and ethanol, respectively, 3 times, followed by drying in a vacuum oven for 8 h at 80 °C. The resulting powder was referred to as UiO-66-F_4_.

### 2.4. Removal Experiments

First 4.0 mL *N*-Ethylmorpholine and 13 mg as-prepared UiO-66-F_4_ were added to a centrifugal tube (50 mL) and ultrasonicated for 5 min till the UiO-66-F_4_ uniformly dispersed in the solution. Then, the tube was placed on a magnetic stirrer with a working speed of 600 rpm. Then, 16 µL of DMNP was injected in the mixture by a microinjector and the stirring speed was maintained. After certain time interval, the MOF particles were separated by a filter, then, 20 µL of the mixture was taken out and diluted to 10 mL to test the concentration of the DMNP. The time intervals were set as 0, 1, 3, 5, 10, 20, 30, 40, 60, and 100 min. When testing the piezoelectric degradation efficiency, the stirring was replaced by sonication with a frequency of 45 kHz. As the DMNP was decomposed into p-nitrophenol, the concentration of the DMNP was inferred by the concentration of the p-nitrophenol which can be detected by UV-Vis with characteristic wavelength of 407 nm.

### 2.5. Calculation Method

The calculations were performed using the first-principle calculation implementation of CASTEP (Materials Studio 2020) [40]. The generalized gradient approximation (GGA) with the Perdew–Burke–Ernzerhof (PBE) formula was employed for the exchange–correlation potential [41,42]. The Broyden–Fletcher–Goldfarb–Shanno (BFGS) method was used to search for the ground state of the supercells, and the convergence tolerance was set to energy change below 10^−5^ eV per atom, force less than 0.02 eV Å^−1^, stress less than 0.05 GPa, and displacement change less than 0.001 Å. The cutoff energy of the atomic wave functions was set to 450 eV.

## 3. Results

### 3.1. Assembly and Characterizations

After 4 h of incubation through microwave-assisted heating, two-dimensional UiO-66-F_4_ nanosheets were successfully obtained. Firstly, the morphology of the as-prepared sample was examined through the utilization of SEM, TEM, and AFM techniques. The SEM image demonstrates the sample’s remarkable homogeneity (Figure 1a), while the enlarged SEM image reveals that nearly all UiO-66-F_4_ clusters comprise interlaced nanosheets measuring approximately 200 nm in size (Figure 1b). The TEM image provides stronger evidence for this conclusion (Figure 1c). Furthermore, both the TEM image and the high-resolution TEM image reveal that the nanosheets appear transparent under fluorescent irradiation, indicating their ultrathin nature. However, upon observation using high-resolution TEM (Figure 1d), the crystal lattice was not clearly visible, indicating a relatively small crystal size. Consequently, this led to inferior crystallinity, which aligns with numerous reported UiO-66-type MOFs [43,44]. In order to further investigate the composition of the sample, the distribution of corresponding elements within a specific area was measured. The results exhibit a remarkable coincidence in the distributions of C, O, F, and Zr (Figure 1e,i), indicating a coordinated interaction between the Zr clusters and ligands. To accurately measure the thickness of the nanosheets, an atomic force microscope (AFM) was utilized. The results indicate that the sample is composed of multiple nanosheets, as depicted in Figure 1j. Subsequently, a specific region was selected for thickness measurement of the nanosheets, indicated by the white line in Figure 1j. The resulting height data reveal that the thickness of these nanosheets is approximately 5 nm, as shown in Figure 1k.

To further illustrate the compacted structure, a collection of size distributions was gathered. The results reveal that the peak on the curve begins at approximately 200 nm and attains its maximum at around 500 nm, indicating that the measured average size was approximately 500 nm (Figure 2a). This aggregation is primarily attributed to the remarkable tendency of the nanomaterial to aggregate, with multiple nanosheets coalescing into larger particles. When measuring, the equipment was capable of detecting only the particles and not single petals. Additionally, the zeta potential reveals that the weighted average potential is approximately 34.2 mV, indicating the nanosheet’s remarkable hydrophilicity (Figure 2b). The porous structure was examined through the N_2_ adsorption–desorption curve and the distribution of pore sizes, as depicted in Figure 2c. The results indicate that the specific surface area is as impressive as 331.827 m^2^/g, aligning with our previously reported findings [38,44]. Furthermore, the weighted average pore diameter has been measured at 0.9 nm, as depicted in Figure 2d. To investigate the crystal structure of the sample, an XRD analysis was performed. The results demonstrate that the planes of the synthesized UiO-66-F_4_ align well with the simulated data (Figure 2e). Notably, the plane of (002) merges with the plane of (111), resulting in a single crystal plane. This observation suggests that the crystals in UiO-66-F_4_ exhibit a uniform crystal plane orientation. The XRD pattern further elucidates the disparities between UiO-66-F_4_ synthesized in this study and those reported in numerous prior works [38]. To investigate the coordination mode of the carboxylic acid group, FTIR spectra were collected for the ligands of UiO-66-F_4_, as carboxylate can exhibit distinct coordination modes that would result in variations in the corresponding spectrum. The broad peak from 2000–3200 cm^−1^ of the ligand is ascribed to the O−H and C=O stretching modes, and it disappears due to the full deprotonation and reduction of the ligand after the coordination process [45]. Moreover, the FTIR spectra of UiO-66-F_4_ samples reveal two characteristic peaks located at around 1637 and 1407 cm^−1^, which correspond to the C=O symmetric and asymmetric stretching vibration. The two peaks with Δ > 200 cm^−1^ (***ν***_as_ − ***ν***_s_ = 230 cm^−1^, Figure 2f) indicate that the carboxylate ligands adopted a bridging bidentate coordination mode [46,47].

To gain a deeper understanding of the coordination mode, XPS analysis was employed to investigate the valence states of the elements within the ligands and metal clusters, as depicted in Appendix A. The peak of the C species can be resolved into three distinct peaks, located at 284.7, 287.2, and 288.8 eV (Figure 3a), which are ascribed to C-C/C=C, C=O, and C-C=O, respectively [48,49]. The O atoms serve as a crucial constituent of clusters, with the deconvoluted peak of O 1s assigned to specific energy levels of 530.3, 531.6, and 533.2 eV, which are ascribed to C=O, O-metal (O-Zr), and O vacancy (Figure 3b), respectively [50,51]. And a distinct peak appears at 687.2 eV on the spectrum of F 1s (Figure 3c), which is attributed to the presence of C-F bonds, thus demonstrating the F did not attend the coordination. The Zr 3d spectra exhibits a spin–orbit doublet that splits into 182.5 and 184.5 eV, which is ascribed to 3d_5/2_ and 3d_3/2_ (Figure 3d) [43,52], respectively, strongly demonstrating the formation of [Zr_6_O_4_(OH)_4_(–COO)_12_] in the frameworks [38,53,54,55].

Based on the aforementioned results, a putative coordination pathway can be inferred. BDC-F_4_ is a is a strictly symmetric dicarboxylic acid, with two carboxyl groups situated on opposite sides of the benzene ring, thereby offering two coordination sites. The FTIR and XPS spectra obtained after the assembly process unambiguously reveal the presence of a bridging bidentate coordination mode between the ligand and the [Zr_6_O_4_(OH)_4_(–COO)_12_]. Utilizing these findings, we propose a mechanism for the formation of 2D UiO-66-F_4_ nanosheets, which is schematically illustrated in Figure 4. Utilizing microwave heating at 100 °C, the carboxyl group of BDC-F_4_ gradually interacted with Zr ions, as depicted in Figure 4a. This interaction led to the formation of [Zr_6_O_4_(OH)_4_(–COO)_12_] clusters, which are illustrated in Figure 4b. Subsequently, these clusters and ligands combined periodically to develop a network structure, shown in Figure 4c. Finally, through the addition of regulators and solvents to unsaturated coordination sites, the network was compacted and folded into 2D nanosheets, as seen in Figure 1b. Conventionally, the synthesis of UiO-66-type MOFs via hydrothermal methods entails prolonged reaction durations exceeding four days. However, our approach utilizes microwave irradiation to rapidly precipitate Zr ions with ligands, favoring the formation of nanosheets. This method offers a straightforward and remarkably swift pathway for the large-scale production of 2D UiO-66-F_4_ MOFs.

### 3.2. Mechanically Enhanced Detoxification Performance

The as-prepared UiO-66 nanosheets were introduced to detoxify the CWA simulant. Firstly, we investigated the mechanical property of the fabricated nanosheets by piezoresponse force microscopy (PFM). The phase hysteresis loop and amplitude loop were measured to evaluate the polarization degree of the MOF crystals. As reported, for the two hysteresis loops, the lower the overlap between the blue and red lines, the higher the degree of MOF crystal polarization [56,57]. As shown in Figure 5a, the sample exhibits well-defined 180° phase-reversal hysteresis, demonstrating a characteristic polarization switching behavior of the as-prepared MOF. Furthermore, a typical amplitude–voltage butterfly loop is obtained under a 10 V DC bias field (Figure 5b). The larger amplitude displacement of UiO-66-F_4_ (~47 mV) indicates a stronger piezoelectric response than that of most reported UiO-66-type MOFs. Herein, the introduction of the F_4_-BDC^2−^ linker that coordinated weakly to the Zr(IV) metal centers might have caused a strong polarity in the UiO-66-type MOFs. In addition, the strong hydrogen bonding between F and μ_3_-OH can also lead to a large polarization of the nanosheets [37,38,58].

Subsequently, we subjected the fabricated MOF to CWA detoxification. Given the severe toxicity of sarin (as demonstrated in Appendix A) and the necessity to conduct relevant experiments in high-standard laboratories, we chose DMNP as a representative of nerve agent simulants. As reported, DMNP can undergo hydrolysis to form p-Nitrophenol. Consequently, the detoxification performance of the prepared MOF can be evaluated by monitoring the concentration change in p-Nitrophenol [1]. The results indicate that, in a sonic environment, the conversion remains relatively unchanged after 100 min, indicating that the DMNP lacks self-degradation capabilities under stirring or ultrasonic conditions (Figure 6e). When UiO-66-F_4_ is added to the catalytic system, the UV-Vis characteristic peak of DMNP gradually diminishes, while the peak of *p*-Nitrophenol concurrently increases (as shown in Figure 6a) under stirring conditions, which indicates that DMNP is efficiently hydrolyzed into *p*-Nitrophenol. In sonic conditions, UiO-66-F_4_ demonstrates a superior hydrolysis efficiency compared to stirring conditions. Furthermore, the hydrolysis model aligns more closely with first-order reaction kinetics [59]. To more clearly investigate the hydrolysis performance under varying conditions, the introduction of the reaction rate constant, k, is employed [44]. The values are calculated as 0.040 and 0.058 min^−1^ for stirring and sonic conditions (Figure 6b,d), respectively, and the *k* is upgraded 45% by sonication with the addition of the UiO-66-F_4_. Furthermore, the half-life time has been reduced from 13.61 min to 10.58 min, as demonstrated in Figure 6f. This significant reduction further underscores the ability of sonication to enhance the hydrolysis performance of UiO-66-F_4_. ZnO is a prototypical inorganic material that exhibits remarkable piezocatalysis properties across diverse fields, making it a highly versatile and responsive material [60,61]. To further validate the distinct piezocatalysis characteristics of the UiO-66-F_4_ material prepared, ZnO was incorporated to assess the hydrolysis performance of alternative piezoresponse materials. However, the findings reveal that ZnO exhibits minimal catalytic activity for the hydrolysis of DMNP, even under sonication conditions (Figure 6e). Notably, the half-life of DMNP extends to 1487 min, indicating that conventional piezoelectric materials are ineffective at converting mechanical energy into chemical energy during the removal of DMNP. These results demonstrate that DMNP can be efficiently removed by piezoresponsive MOFs, aided by mechanical vibration. Additionally, the detoxification performance of UiO-66-F_4_ was compared with various reported materials, revealing that the approach utilized in this study exhibits comparable detoxification efficiency (Table 1).

To gain a deeper comprehension of the alterations that occur under mechanical stress, a theoretical calculation was undertaken to emulate the responses exhibited by MOF crystals. The results initially reveal that the crystals maintain their integrity even when subjected to significant mechanical stress (Figure 7), indicating that MOF crystals possess remarkable stability, even under extreme conditions. Furthermore, the Z projection and side view of the MOFs’ crystal structure reveal significant deformation. Specifically, the crystal has shrunk by approximately 5% in the direction of applied stress. Nevertheless, the metal cluster and the benzene ring of the ligand retain rigidity, resisting distortion. This deformation is primarily attributed to the rotation of carboxyl groups on the ligands’ benzene rings, as evident from the side view of the crystal structure.

Moreover, the band gap structure determined the generation of radical oxidation species (ROS) [68,69,70]. We also investigated the band gap changes in the MOFs under stress (Figure 8a,b). The results indicate that the band gap is 2.313 eV, which aligns closely with our prior research [38]. Upon the application of mechanical stress, the compressive strain notably altered the band gap structure, resulting in an increase to 2.405 eV. Simultaneously, the bands in the valence zone exhibit a significant increase under stress, as illustrated in Figure 8b. This observation underscores the fact that the band gap structure can be effectively modulated by mechanical stress [71,72]. The density of states (DOS) analysis offered deeper understanding of the distinct bands’ origins, as illustrated in Figure 8c. Notably, the application of mechanical stress caused the peak value at the Fermi level to increase. Concurrently, the conduction band shifted slightly towards the high-energy level, leading to an expansion in the band gap. This observation aligns well with the calculated band gap results. The CB primarily aligns with the Zr 3d electrons or with orbitals that are delocalized throughout the entire linker (as shown in Appendix A), demonstrating a relatively low sensitivity to the functional group. Moreover, the application of mechanical stress causes the polarization to increase at an accelerated rate. Consequently, a novel VB’ emerges, significantly broadening the band gap (Figure 9b).

Drawing upon the aforementioned results and preceding research, a preliminary band diagram can be established (Figure 9). During the assembly process of UiO-66-F_4_, the absence or unsaturated coordination of Zr-O clusters can lead to a significant number of metal node or ligand defects, ultimately endowing UiO-66-F_4_ with remarkable catalytic capabilities, even without any external assistance. When the UiO-66-F_4_ nanosheets are employed for the removal of CWA simulants, the unsaturated coordination sites can directly react with the ester group of DMNP, thereby achieving the aim of detoxification [1]. After the application of mechanical vibration, the band gap of the nanocrystals widens significantly, leading to the polarization effect. Consequently, a substantial number of free charges (e^−^) are generated on the catalyst surface, efficiently reacting with O_2_ to form $O_{2}^{•-}$. On the one hand, partial $O_{2}^{•-}$ can target the phosphoester bonds (O-P), thereby facilitating the hydrolysis of DMNP as depicted in Figure 9c. On the other hand, $O_{2}^{•-}$ can expedite the generation of additional ^1^O_2_ through interaction with residual holes (h^+^), which can further oxidatively degrade the hydrolysate [73,74,75,76,77].

## 4. Conclusions

In this study, we successfully fabricated two-dimensional UiO-66-F_4_ nanosheets using a microwave-assisted method. These nanosheets exhibit an ultrathin thickness of approximately 5 nm and possess a large specific surface area of 331.827 m^2^/g. These nanosheets possess remarkable mechanical-energy-harvesting capabilities and have the potential to be utilized in the mechanically enhanced detoxification of CWA simulant DMNP. Under optimal conditions, the half-life of DMNP is 10.58 min, and mechanical vibration can significantly enhance its removal performance by approximately 45%. The first-principle simulation unveils the structural transformations within MOF crystals. The mechanical stress potentially influences the rotation angle of the carboxyl groups within the tetrafluoroterephthalic acid, potentially leading to slight deformations along the direction of pressure. Subsequently, this could broaden the band gap and reinstate the band gap structure of the crystals. All the stress-induced changes are advantageous for enhancing the detoxification capacity of UiO-66-F_4_, thereby illustrating the efficacy of mechanical vibration in detoxifying CWAs. This study offers a novel perspective for defending against CWAs through innovative approaches.

## Figures and Tables

**Figure 1 nanomaterials-14-00559-f001:**
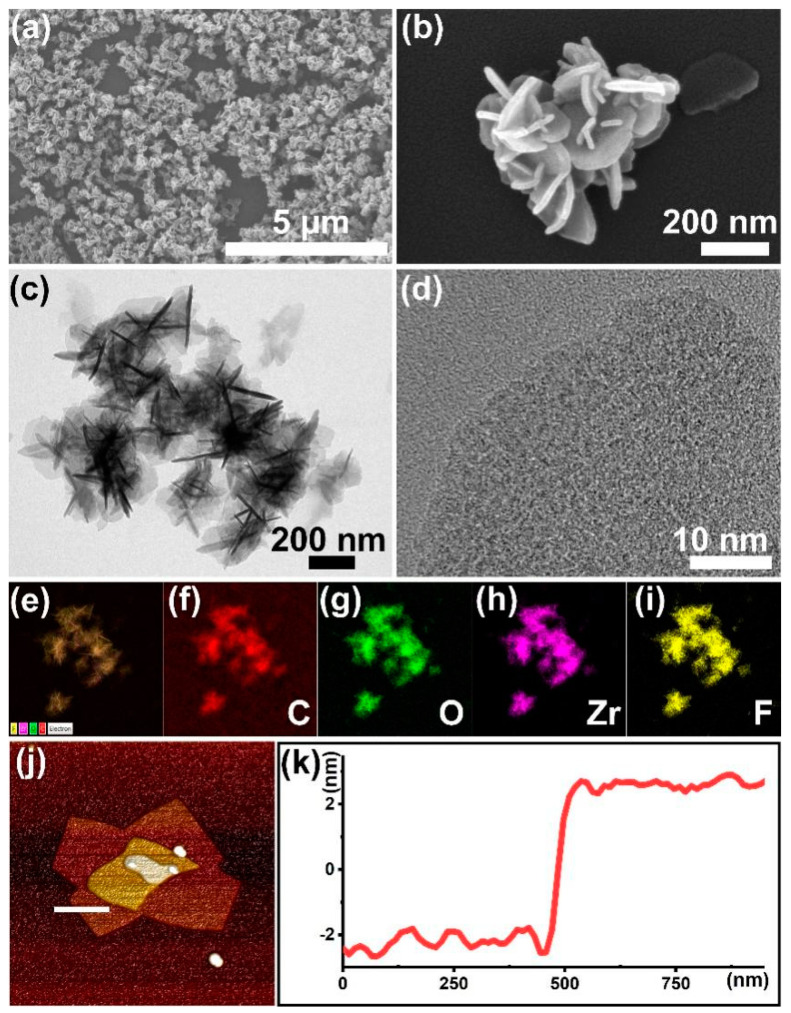
Characterizations of the UiO-66-F_4_: (**a**,**b**) SEM and enlarged SEM images; (**c**,**d**) TEM and HR-TEM images; (**e**–**i**) corresponding element distribution images; (**j**) AFM image; and (**k**) corresponding height data.

**Figure 2 nanomaterials-14-00559-f002:**
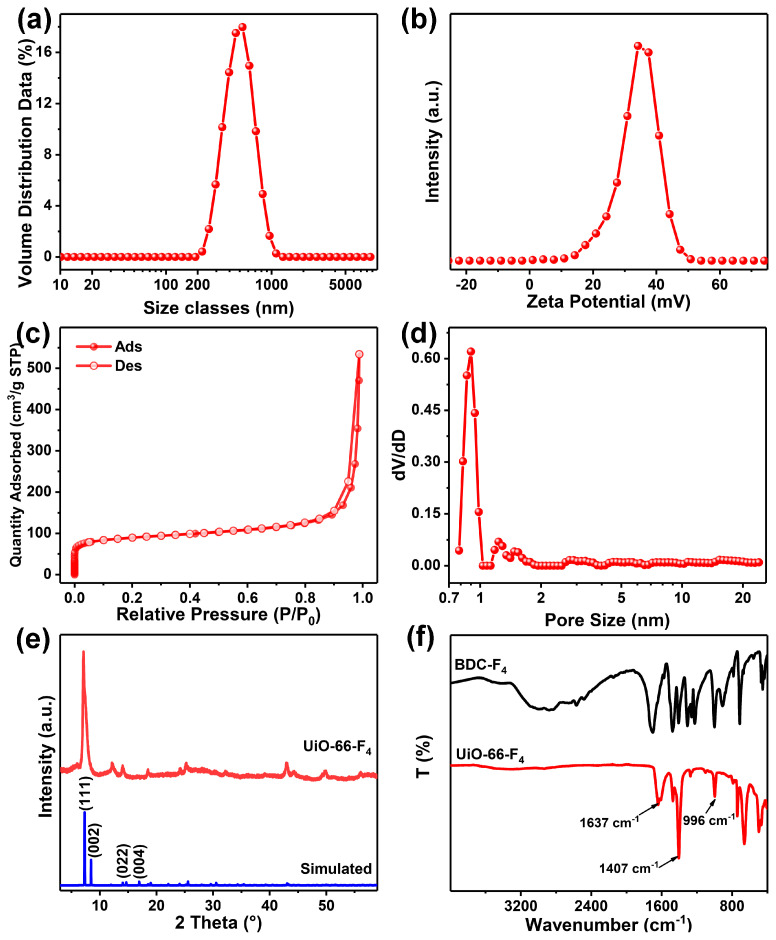
(**a**) The size distribution; (**b**) the zeta potential; (**c**) the N_2_ adsorption and desorption curve; (**d**) the pore size distribution of the UiO-66-F_4_; (**e**) the XRD pattern and the simulated data; (**f**) the FTIR spectra of BDC-F_4_ and UiO-66-F_4_.

**Figure 3 nanomaterials-14-00559-f003:**
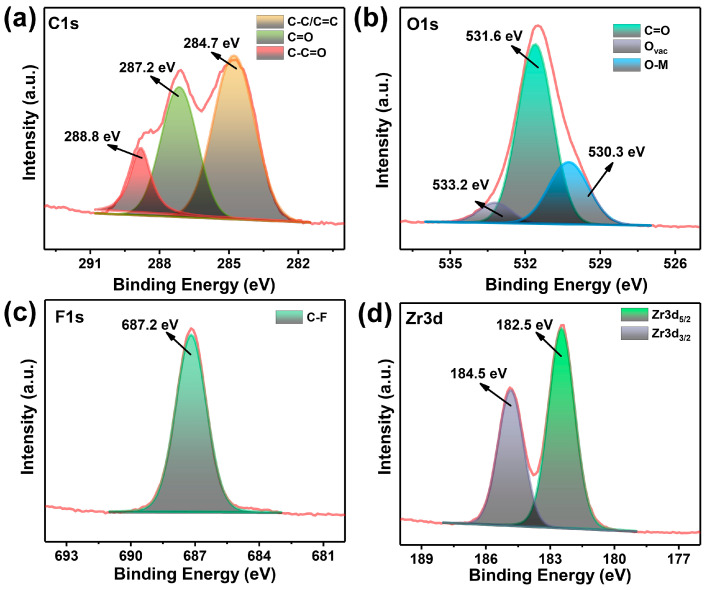
The high-resolution XPS spectra of UiO-66-F_4_: (**a**) C 1s, (**b**) O 1s, (**c**) F 1s, and (**d**) Zr 3d.

**Figure 4 nanomaterials-14-00559-f004:**
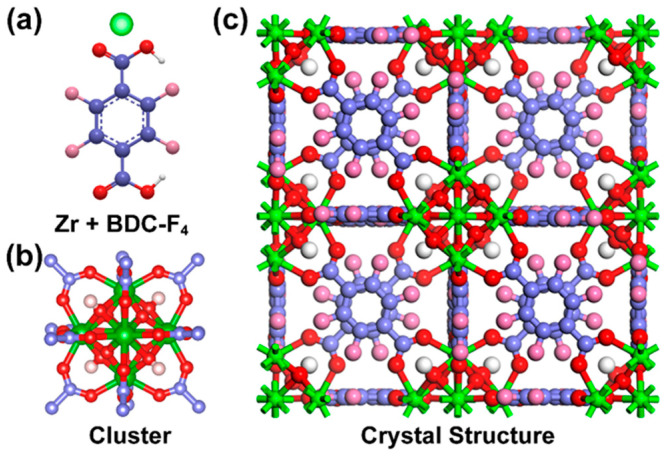
The structure of (**a**) Zr and BDC-F_4_, (**b**) Zr-O cluster, and (**c**) crystal structure.

**Figure 5 nanomaterials-14-00559-f005:**
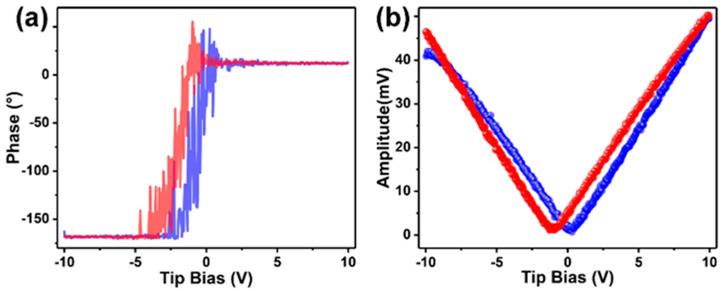
The piezoresponse capacity of the UiO-66-F_4_: (**a**) phase hysteresis loop, (**b**) amplitude hysteresis loop. (Blue line: the change in phase or amplitude with the tip bias voltage gradually increasing; red line: the change in phase or amplitude with the tip bias voltage gradually reducing).

**Figure 6 nanomaterials-14-00559-f006:**
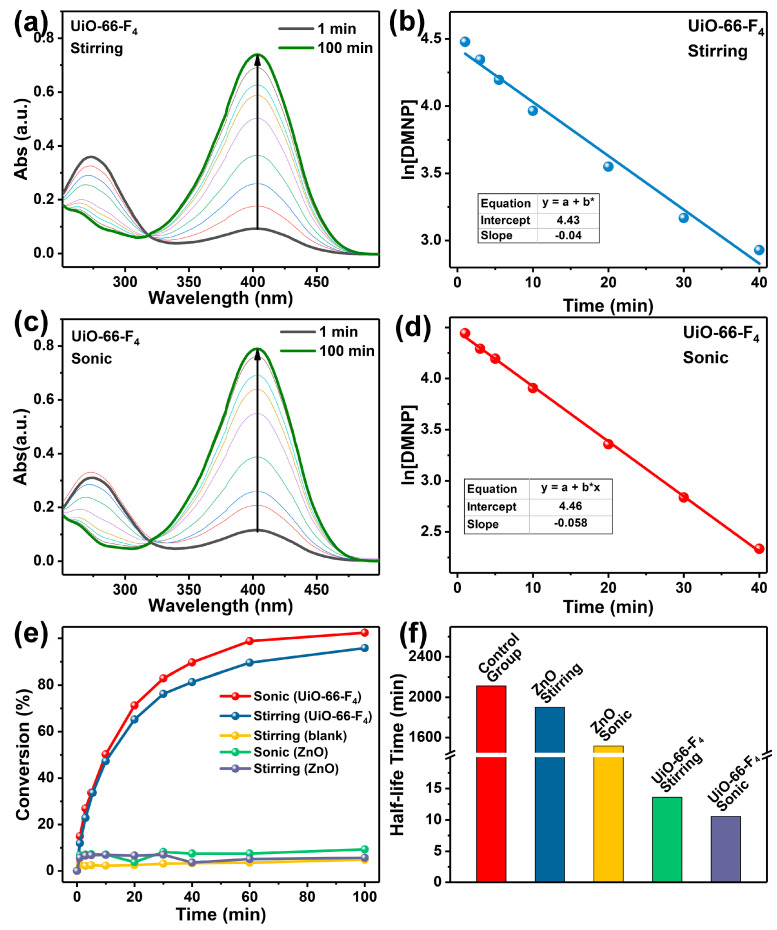
The detoxification experiments results: (**a**) degradation results and (**b**) first-pseudo-order kinetics under stirring conditions; (**c**) degradation results and (**d**) first-pseudo-order kinetics under sonication conditions; (**e**) comparison of degradation performance and (**f**) half-life time results of other control piezoresponse materials.

**Figure 7 nanomaterials-14-00559-f007:**
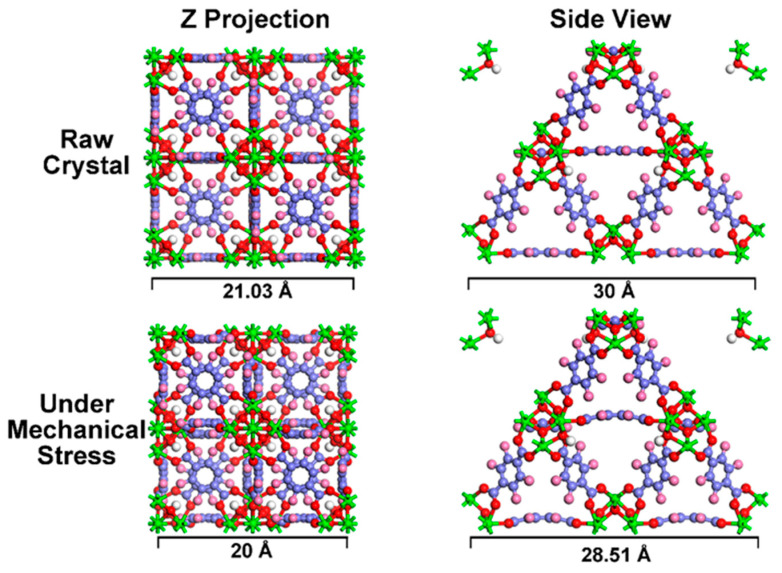
The first-principles calculation implementation of the crystal changes under stress.

**Figure 8 nanomaterials-14-00559-f008:**
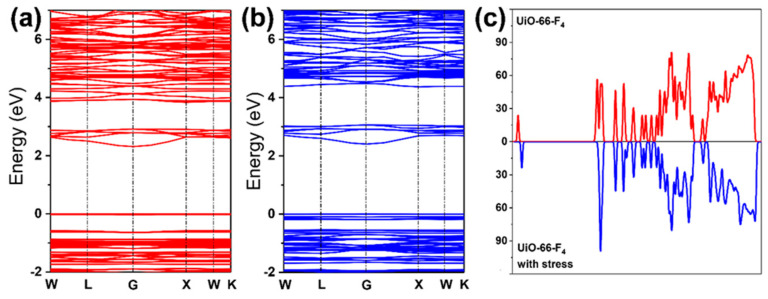
The band gap structure of (**a**) raw UiO-66-F_4_ (2.313 eV) and (**b**) under stress (2.405 eV); (**c**) The DOS of UiO-66-F_4_ and under stress.

**Figure 9 nanomaterials-14-00559-f009:**
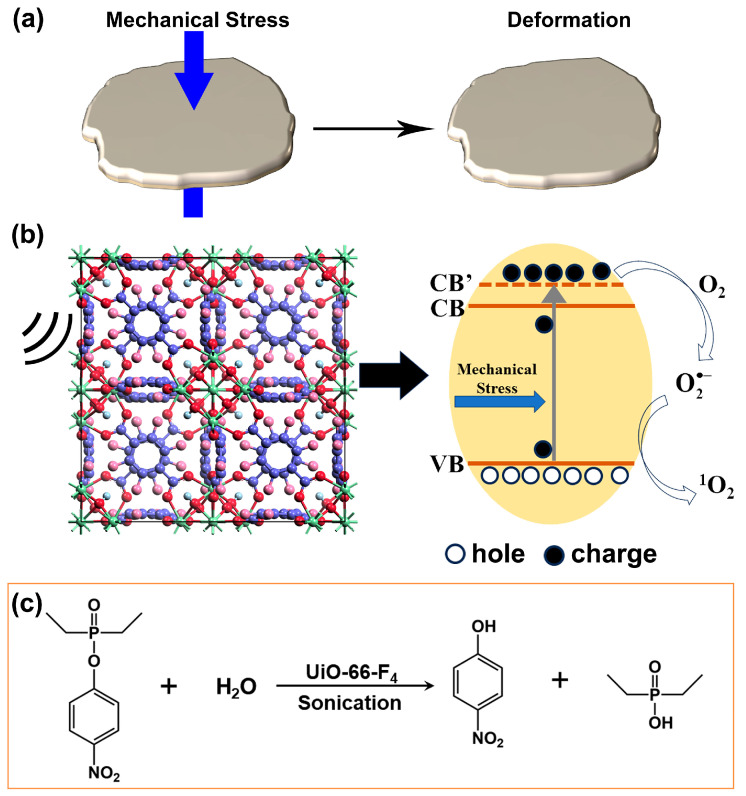
Schematic illustration of (**a**) mechanical deformation of UiO-66-F_4_; (**b**) band diagram; (**c**) detoxification mechanism.

**Table 1 nanomaterials-14-00559-t001:** The comparison with the reported materials.

Catalysts	*t*_1/2_ (min)	*k* (min^−1^)	Ref.
PP/TiO_2_/UiO-66-NH_2_	15	--	[62]
Zr(OH)_4_@PIM-1-Coat	12.6	0.055	[63]
UiO-66-NH_2_/PAN	10	--	[64]
Ce-BDC	8.0	0.087	[65]
MOF@PDMAEA@LiCl@PNIPAM	>60	--	[66]
MIP/UiO-66-NH_2_-0.5	9.4	--	[67]
UiO-66-F_4_	10.58	0.058	This work

## Data Availability

Dataset available on request from the authors.

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
