# Peer review of "Mechanically Enhanced Detoxification of Chemical Warfare Agent Simulants by a Two-Dimensional Piezoresponsive Metal–Organic Framework"

_nanomaterials, 2024, doi:10.3390/nano14070559_

Round 1

Reviewer 1 Report

Comments and Suggestions for Authors

This work is devoted to Metal-organic Framework such as a two-dimensional piezoresponsive UiO-66-F4 exhibited great potential in protection against CWAs in simulants detoxification under mechanical and sonic vibration. Research on this topic is very relevant and perspective, since public protection against CWAs is one the most important issue for people safety. The article is well written and interesting for broad range of readers and the results obtained probably would have the promising application. However, the quality of the text and the figures is not suitable for publication. In principle, article can be published after addressing the following points:

  1. There are no description, chemical or structural formula of DMNP;
  2. Why DMNP was chosen to study?
  3. Page 4, Line 146: References 41,42 should be is superscript
  4. Figure 1(k) looks confusing. Please, mention in details how it was obtained.
  5. Figure 2: the letters (e) and (f) are in reverse order;
  6. Page 5 Lines 163-166: Authors mentioned that the “size of the nanosheet is ~200 nm”, whereas the size distribution at the Fig. 2a clearly shows that average particle size is about 500 nm (the volume distribution data peaks at 500 nm). Please, correct the average size or discuss in details the difference.
  7. Page 5, line 168: the misprint in the word “hydrophily”. Authors should use the term "hydrophilicity"
  8. Page 5, line 174: Authors wrote «And the plane of (002) is fused to the plane (111), which is strongly corresponded to the two-dimensional structure differ from the mostly reported octahedron» This sentence looks confusing. Authors should re-write this sentence to make clear the statement.
  9. Fig 2e: Authors should explain the disappearance of the broadened band of in the region of 2400-3200 cm-1 in UiO-66-F4 comparing to BDC-F4;
  10. p.6, Line 190 — “Zr8+” looks like misprint. Does 8-valent zirconium exist?
  11. Figure 3: letter (с) is repeated twice (last panel should be named as (d));
  12. Figure 5: what does the blue and red lines correspond to?;
  13. There is no explanation of the reason of resulted differences in detoxification experiment (between ZnO and UiO-66-F4). Why DMNP could be more efficiently hydrolysis to p-Nitrophenol than ZnO?
  14. It is mentioned at page 8 (lines 235-246) that detoxification realize due to hydrolysis, but at page 10-11 (lines 297-306) detoxification occurs through oxidation;
  15. On pages 9-11 (where calculations were carried out), it was shown that the increase in the band gap is associated DMNP with the oxidation process. However, as a result of the experiment, the authors reported that detoxification occurs due to hydrolysis. How the presented calculations is related to the experimental data obtained?
  16. The choice of calculation methods is not justified;
  17. Conclusion section should be significantly extended. The importance of the performed study should be clearly stated in the conclusions.

Reviewer 2 Report

Comments and Suggestions for Authors

-The manuscript is very difficult to follow; the abbreviations must be defined (what is DMNP?).

- The clear procedures and the instruments used to determine the DMNP concentration must be added to the text.

-Why the adsorption experiments were carried out on a small volume of solution (4 mL + 16 microliter), and how the MOF was separated?

- Explain the need to add N-ethylmorpholine to the treatment media.  

- Delete the text in lines 134-136

- How TEM images show outstanding light transmission ability of MOFs (Line 143)

-Please rewrite “Therefore, almost no clear crystal lattice can be observed, that means the crystallinity of the sample is relatively inferior” in clear language.

-How the size distribution was collected? add the analytical methods used to the characterization section.\

- How the two dominant peaks located at around 1637 and 1407 cm-1 indicate that the carboxylate ligands adopted a bridging bidentate coordination mode

-What are Blue and red colors refer to, add to figure 5.

- Figure 6 a, c and e show that sonication lead to higher conversion than stirring, while the Abstract has different opinion, explain.

- What is the benefit of converting toxic analyte to another toxic species (p-nitrophenol).

-The mechanism of the conversion should be explained in detail

-Add the equations of kinetics modelling to the manuscript, also the chemical structures of the analytes.

- Please compere the efficiency of the UiO-66-F4 with other materials in literature.  

Comments on the Quality of English Language

English typos, abbreviations and references numbering in the text must be corrected.

Reviewer 3 Report

Comments and Suggestions for Authors

Nanomaterials 2884677

In this paper, the authors obtained two-dimensional piezoresponsive UiO-66-F4 and subjected in CWAs. The results show the mechanical vibration. The reaction rate constant K was upgrade 45% by sonication.

This paper is interesting, the introduction is good, the references are complete. The investigation is well designed, and the presentation is well, but is necessary that the authors make some changes in the manuscript

Comment

1)      In Abstract, define CWAs and DMNP. Revise all manuscript

2)      Line 123, change UV-vis for UV-Vis. Revise all manuscript

3)      Line 127, change potential.39,40 for potential.39,40 Revise all manuscript

4)      Line 146, change MOFS.41, 42 for MOFS41,42. Revise all manuscript

5)      Revise and rewrite the paragraph “To further investigate….in the frameworks.”

6)      Complete the comments of the IR spectra

7)      Revise presentation of the Figure 5

8)      Revise presentation of the Figure 6

9)      Line 238, Is reference 1?

Comments on the Quality of English Language

Moderate revision of English is necessary

Reviewer 4 Report

Comments and Suggestions for Authors

I revised the manuscript of the article titled “Mechanical-Enhanced Detoxification of Chemical Warfare Agent Simulants by Two-dimensional Piezoresponsive Metal-Organic Framework” submitted to the nanomaterials journal. 

The manuscript presents an interesting idea for research. Degradation of chemical warfare agents with the use of MOF materials is a good idea and an important subject of research at present. The presented research is well planed, the synthesis and investigation of materials were done properly and the catalytic decomposition of the CWA was investigated. I do not find many weaknesses in the subject, except potential real application, when contaminated areas should be cleaned, and the lack of precise mechanism of the catalytic reaction

In my opinion, the paper is worth publishing, and I strongly encourage the Editor to accept the paper.

I attached a list of a few small problems that I encountered while reading the manuscript:

·         Please provide all abbreviation explanation (e.g. DMNP).

·         ``However, mechanical-enhanced detoxification of 66 CWAs by piezoresponsive MOFs has rarely been reported.´´ - citation needed.

·         Please, correct editorial mistakes, e.g.: ``of CASTEP.38 The generalized´´

·         Please unify citations number style.

·         3. Results, line 134-136 should be deleted.

·         Line 186, please describe what is M in O-M.

·         I found lack of important citation doi:10.1039/C5CC08972G.

Round 2

Reviewer 1 Report

Comments and Suggestions for Authors

The authors addressed many comments, but some of them are still required the attention:

Q6 Authors answered this comment in response to reviewer but did not clarify size distribution issue in main text. Please, make changes in the main text.

Q10 I did not find any information about Zr8+ ion in the presented papers (Chem. Mater., 2010, 22, 6632–6640; Inorg. Chem., 2016, 55, 8241–8243). Authors should clarify the possibility of existance 8-valent zirconium exist.

Q12 Authors should add the description of red and blue lines in Figure 5 caption

Reviewer 2 Report

Comments and Suggestions for Authors

Still the authors did not fully address all the comments.

-Why specifically N-ethylmorpholine was added to the treatment media, why not NaOH if it is just to provide an alkaline media as the authors responded?

- Can you please explain how the tiny volume of samples 20 microliters was measured by UV-Vis, does it need a special procedure? Instead why the authors didn’t use diluted solutions in bigger volumes?  

- Also, the authors ignored the mechanism of degradation in the absence of sonication.

- What crystals do the authors mean in line 63?

-still “DMNP” is not defined in the abstract. Also, should be defined when first mentioned in the text, please correct.

Comments on the Quality of English Language

Minor editing of English language required
